# Recent Advances in DNA Nanotechnology for Plasmonic Biosensor Construction

**DOI:** 10.3390/bios12060418

**Published:** 2022-06-15

**Authors:** Jeong Ah Park, Chaima Amri, Yein Kwon, Jin-Ho Lee, Taek Lee

**Affiliations:** 1Department of Chemical Engineering, Kwangwoon University, Seoul 01897, Korea; m3m33@kw.ac.kr (J.A.P.); ijmr9126@kw.ac.kr (Y.K.); 2Department of Convergence Medical Sciences, School of Medicine, Pusan National University, Yangsan 50612, Korea; a.chaima@pusan.ac.kr; 3School of Biomedical Convergence Engineering, Pusan National University, Yangsan 50612, Korea

**Keywords:** biosensors, DNA nanotechnology, DNAzyme, plasmonic biosensor, aptamer, DNA origami

## Abstract

Since 2010, DNA nanotechnology has advanced rapidly, helping overcome limitations in the use of DNA solely as genetic material. DNA nanotechnology has thus helped develop a new method for the construction of biosensors. Among bioprobe materials for biosensors, nucleic acids have shown several advantages. First, it has a complementary sequence for hybridizing the target gene. Second, DNA has various functionalities, such as DNAzymes, DNA junctions or aptamers, because of its unique folded structures with specific sequences. Third, functional groups, such as thiols, amines, or other fluorophores, can easily be introduced into DNA at the 5′ or 3′ end. Finally, DNA can easily be tailored by making junctions or origami structures; these unique structures extend the DNA arm and create a multi-functional bioprobe. Meanwhile, nanomaterials have also been used to advance plasmonic biosensor technologies. Nanomaterials provide various biosensing platforms with high sensitivity and selectivity. Several plasmonic biosensor types have been fabricated, such as surface plasmons, and Raman-based or metal-enhanced biosensors. Introducing DNA nanotechnology to plasmonic biosensors has brought in sight new horizons in the fields of biosensors and nanobiotechnology. This review discusses the recent progress of DNA nanotechnology-based plasmonic biosensors.

## 1. Introduction

Nucleic acids are regarded as the one of the most vital molecules in living organisms; they have been credited with the origin of life on Earth [1,2]. Our understanding of nucleic acids has led to immense advances in medicine, pharmacology, biology, and biotechnology [3,4,5]. In particular, the convergence of biotechnology and nanotechnology has resulted in the development of innovative technologies for biosensors, drug delivery, and bioimaging [6,7,8]. Among these, DNA nanotechnology has received the most attention from researchers interested in the material rather than genetic characteristics of DNA [9,10]. DNA nanotechnology has brought forth a new paradigm for DNA research, one that employs the genetic functionality of DNA for engineering applications in biosensors [11,12], nanoarchitecture [13,14], drug delivery [15,16], and computations [17,18]. This integrated approach has resulted in the development of a tremendous line of products, which would have been impossible with conventional nucleic acid research or nanobiotechnology alone.

Prior to the rise of DNA nanotechnology, DNA was regarded as a molecule studied exclusively in the fields of life sciences and medicine. DNA nanotechnology has shifted its focus to the valuable material characteristics of DNA, which provides unusual stability, flexibility, complementary binding, and tailored functionality [19,20,21]. Several researchers have reported on DNA-based computations [22], DNA origami structures [23], DNA-based biosensors [24], DNA-based vaccines [25], and drug development using DNA [26]. Of note are the DNA aptamers [27,28,29] and DNAzymes (deoxy → ribozymes) [30,31,32] that mimic the characteristics of natural antibodies and enzymes, respectively. A DNA aptamer can bind a specific target with high binding affinity. Usually, the DNA aptamer and DNAzyme can be produced by the systematic evolution of ligands by exponential enrichment (SELEX) that can bind specifically designed targets [33]. An aptamer is nucleic acid or peptide sequence that can mimic the functions of an antibody. The terms aptamer and SELEX were reported by different research groups [34,35].

Several groups reported on the application of aptamers in medicines [28], pharmacology [36], and molecular biology research [27]. Several types of aptamers are reported to have biosensor and diagnostic applications [37,38,39,40,41]. The DNA or RNA aptamer is a great bioreceptor that can alter an antibody with the manufacturing cost. Aptamers can also be produced via chemical synthesis without requiring experiments on animals for antibody production. Moreover, compared to antibodies, chemical production can easily be scaled up in aptamers because of the short production time. DNAzymes (Deoxyribozyme), also known as catalytic DNA, act as enzymes that can catalyze specific reactions [41,42]. These DNAzymes can be produced by the SELEX method [28]. Unlike natural enzyme, DNAzymes were reported to have several functionalities such as DNA self-modification [43], RNA cleavage [44] and H_2_O_2_ reduction [45]. These unique characteristics make DNAzymes a good material for biosensor construction as the bioprobe [43,44,45,46].

In particular, the study of DNA nanotechnology has led to innovative advances in biosensors. Here too, the structural flexibility and malleability of the structure of DNA allows the nucleic acid to be employed as the bioprobe [23,47,48]. For example, the DNA 3-way junction (3WJ) structure has added one more sequence to the typical double-stranded DNA that can add extra functional groups to the structure [49,50]. Likewise, Y-shaped DNA gives a similar effect [51,52]. Based on this concept, DNA 4-way junctions (4WJ) provide additional arms to improve the functionality of biosensor fabrication [38,53]. G-quadruplex DNA-Based Biosensors are a good example of the structural application of DNA nanotechnology [54,55]. Conventional single-stranded DNA can bind complementary sequences; this principle is used as the basis of a PCR-based in vitro diagnostic (IVD) apparatuses or other genetic biosensor constructions. DNA origami, in contrast, provides unprecedented DNA structures that expand the various structural bioprobes. Thus, DNA nanotechnology offers new methods for biosensor development.

Incorporating DNA nanotechnology with the plasmonic biosensors provided several merits compared to conventional protein-based biosensors. (1) The multi-functional bioprobe can minimize the bioprobe construction. For example, the DNA can be easily manufactured by chemical synthesis that means we can easily introduce the chemical linker or fluorescence dye at the end of each terminal. (2) The production cost of aptamer is cheap compared to a protein or antibody. (3) DNA hybridization with nanomaterial provides the new sensing platform for plasmonic biosensor. Based on those advantages of DNA nanotechnology, the focus of this review will be on the recent progress of nucleic acid engineering for plasmonic biosensor construction. The applications of both simple aptameric structures and origami-shaped structures in SPR, SERS, and MEF will be discussed.

Meanwhile, researchers in the field of biosensing have shifted their attention to plasmonic biosensors. This propriety can be harnessed through multiple applications in the field of biosensing [56]. For instance, to generate surface plasmon resonance (SPR), an incident light produces energetic electrons that propagate through the surface [57]. In contrast, in local surface Plasmon resonance (LSPR), coherent oscillations are confined to a metallic nanostructure and cannot propagate thus generating an electromagnetic field (EM) [58]. These properties can be harnessed for multiple applications in the field of biosensing [56]. For instance, when a molecule, such as a protein or an aptamer, is adsorbed on to the plasmonic surface, the refractive index changes and such values can be measured. SPR/LSPR is a highly sensitive, simple, low cost, and label-free detection method [59,60]. Raman scattering is the inelastic scattering of photons caused by a high-intensity incident light. This effect generates a spectrum fingerprint proper to the molecule of interest [61]. However, such measurements are not precise at low concentrations or for complex structures such as proteins [62]. To overcome this issue, surface-enhanced Raman scattering (SERS) combines a major LSPR EM-induced enhancement and a lesser chemical enhancement induced by Raman reporters. Thus, leading to a stronger peak detection signal in SERS that can provide information on the chemical species and structure of a wide range of molecules [63]. Metal-enhanced fluorescence (MEF), also referred to as radiative decay engineering, is another application of LSPR EM that has been extensively studied. It is known that radiating surface plasmons can affect the spectral properties of a fluorophore [64], i.e., the combination of a metal and a fluorescent molecule at optimal distance has been shown to increase fluorescence emission, prevent photobleaching, and increase the sensitivity of detection, among other things [65].

As the field of nucleic acid-based Plasmon biosensors is continuously evolving, a regular reviewing of the latest breakthroughs and the most trending technologies is necessary. To highlight the recent progress of nucleic acid engineering for Plasmonic biosensor construction, this review will be chiefly focused on research outcomes published within the last five years. The applications of both simple aptameric structures and origami-shaped structures in SPR/LSPR, SERS, and MEF will be discussed.

## 2. Aptameric Structure-Based Plasmon Biosensor

Despite the promising results shown by antibody-bound plasmon-generating structures in sensing a multitude of molecules from proteins to whole cells, the biosensor trend is progressively shifting toward aptameric detection probes for numerous reasons. Not only do aptasensors have a low cost of production and are easy to use, but they also have repeatedly demonstrated a high sensitivity to an extensive range of targets.

### 2.1. SPR/LSPR-Based Sensing

Using broad-spectrum antibiotics can significantly affect both the environment and the living organisms involuntarily exposed to them by generating drug resistant bacteria. Oxytetracycline (OTC), a semi-synthetic antibiotic, is commonly used in animal feeds and is often released into the environment. In Huang et al.’s experiment, a liquid crystal catalytic amplification-nanogold SPR aptamer absorption assay was developed for the detection of trace OTC [66]. The team developed a novel SPR spectrophotometric analysis method by combining 4-heptylbenzoic acid (HPB) and HAuCl_4_. The reduction of HAuCl_4_ to gold NPs was hindered by the adsorbed OTC-binding aptamers. When in the presence of the target molecule, the aptamer would be released and the NPs would catalyze the reaction. This sensing mechanism achieved a detection limit of 0.50 ng/mL for OTC [66]. In their research, Écija-Arenas et al., developed an aptamer-based SPR biosensor for the determination of kanamycin, an aminoglycoside bacteriocidal antibiotic, residue in food [67]. The team tested the combination of a gold and a graphene film through two methods: self-assembly of reduced-graphene (rGO) and chemical vapor deposition (CVD) of graphene (GO) (Figure 1a). The former step consisted of a 1-pyrenebutyric acid (PBA) modification via π-stacking before binding the kanamycin specific aptamer to the graphene surface. As a proof of concept, the sensor was tested on kanamycin spiked commercial cow milk samples. Overall, the GO-based sensor had a detection limit of 285 nmol L^−1^, seven-fold lower than the rGO-based platform. This was most likely due to the homogeneity of the GO monolayer when directly deposited through CVD [67]. As the sensitivity of SPR sensors can be affected by the homogeneity of the probe surface, producing a homogeneous deposition is one of the challenges to overcome. In their study, Chen et al., constructed an SPR aptasensor for the detection of SARS-CoV-2 based on a gold chip coated with thiol-modified niobium carbide MXene quantum dots (Nb2C-SH QDs) [68]. This was followed by the immobilization of a detection aptamer (N58) through π-π stacking, electrostatic adsorption and hydrogen bond on to the chip. The multifunctional proprieties of the chosen nanoparticles allowed for a homogeneous deposition leading to a detection limit of 4.9 pg/mL for the N-gene of SARS-CoV-2. Furthermore, the sensor displayed good selectivity in the presence of other viruses and a great stability when tested in other environments such as human serum [68].

Arsenic is a naturally occurring poisonous non-metallic element. The standard method, silver diethyldithiocarbamate spectrophotometry (SDDC), for determining the arsenic concentration in water is simple but has a detection limit of 0.5 mg/L. Considering that ingesting 1 mg of arsenic is highly toxic for the human body [69,70], the development of a more sensitive detection method is necessary. Through a catalytic amplification method based on colored nanosilver surface plasmon resonance (SPR), Zhang et al., achieved a detection limit of 0.01 μg/L [69]. The sensing mechanism was based on the catalytic effect of gold-doped carbon dots (CDAu) in the reductive reaction of AgNO_3_ with glucose. In the initial stage, aptamers were adsorbed on to the CDAu surface, stopping the catalysis. Only in the presence of arsenic (As^3+^), which forms a conjugate with the aptamer, would the carbon dots be released. The ensuing reaction would lead to the synthesis of yellow spherical silver nanosols exhibiting an intense SPR absorption peak (Figure 1b) [69].

Both SPR and LSPR aptameric sensors offer great advantages when compared to other sensing platforms. However, due to the nature of the interaction between the aptamer, the target, and the LSPR/SPR generating structure, certain factors must be taken into account. Some detection probes are solely based on the adsorption of a target detection aptamer onto a metallic nanostructure which in the presence of the target would detach and subsequently modulate the aggregation of the nanoparticles. Yet, in certain cases, the target itself has been shown to adsorb onto the nanostructures. For instance, Ochratoxin A (OTA), one of the most common food contaminants, an extremely toxic molecule for living organisms, can also adsorb onto AuNPs, thus making accurate detection difficult [71,72]. This phenomenon was first described by Liu et al., during their attempt to widen the detection range of OTA by doubling the calibration curve of their gold NP aptamer-based localized surface plasmon resonance (LSPR) sensor [71]. The team established that at low concentration, OTA would bind to its aptamer inducing the aggregation of the AuNPs but as the concentration increases the free OTA would directly bind to the AuNPs and protect them from aggregating. By developing a double calibration curve the sensing mechanism achieved a detection limit of 10^−10.5^ g/mL. More importantly, the same effect was observed for different analytes such as adenosine triphosphate (ATP) and 17β-estradiol (EST) [71].

### 2.2. Surface-Enhanced Raman Scattering-Based Sensing

Herbicides have detrimental side effects on human health and their long term exposure can lead to chronic disease and childbirth defects [73]. To achieve the detection of glyphosate (GLY) at very low concentrations, Liu et al., constructed a simple and sensitive quantitative analysis based on gold nanoplasmon resonance Rayleigh scattering (RRS) and SERS [74]. In this study, gold nanoparticles (AuNPs) were generated through the catalytic effect of a covalent organic framework (COF) on a glycol and chloroauric acid reaction. The COF was prepared by the polycondensation of melamine and p-benzaldehyde, and could interact with a GLY-binding aptamer through hydrogen bonds (HB), hydrophobic interactions (HI), and intermolecular forces (IF). However, in the presence of GLY, the aptamers would detach from the CPF and allow for the catalysis of AuNP synthesis in a proportional manner (Figure 2a,b). The detection of GLY was possible through RRS, but as the blank was high, SERS analysis showed a better signal with a peak at 1617 cm^−1^ using Victoria blue B molecular probes. The team was able to achieve a detection limit of 0.002 nmol/L for GLY [74].

Urea, a protein decomposition product, is another harmful molecule that needs to be monitored [75]. Yao et al., used a similar RRS/SERS dual-mode method to detect urea concentrations as low as 0.06 nM and 0.03 nM, respectively, in milk [76]. The slow reaction of Citrate-Ag (I) can be catalyzed using a polystyrene nanosphere (PN) to produce yellow silver NPs (AgNPs). Similar to the previous GLY detection mechanism, PN adsorbed urea-binding aptamers would weaken the RRS and SERS signal (Figure 2c,d). When in the presence of their target, the aptamers would release the catalytic NPs in a linear manner, thus generating a stronger signal [76].

Cancer detection is also a great application of SERS-based plasmonic sensors due to the low concentration of most biomarkers. In a study by Bhamidipati et al., the combination of gold nanostarts and truncated aptamers allowed for the quantification of epithelial cell adhesion molecules (EpCAM), a common cancer biomarker, at the single cell level with concentrations in the subnanomolar range [77]. As the expression of cancer markers can vary at the cellular level, the quantification of EpCAM density in individual cells can help clinicians monitor the phenotypic evolution of cancer cells. Herein, the biomarker expression of both MCF-7 and PC-3 cancer cell lines, with high and low EpCAM expression, respectively, was successfully measured with aptamer-functionalized gold nanostarts (Figure 3a,b). Interestingly, when compared to a longer, 48-base-pair oligonucleotide, a truncated EpCAM aptamer seemed to have better sensitivity, most likely due to a reduction in the number of conformations possible at room temperature [77]. Ning et al., exploited an aptamer-bound gold–silver bimetallic nanotrepang structure in their SERS analysis for the simultaneous detection of multiple cancer-related exosomes [78]. By alternating between a gold nanorod (GNR) core, Raman reports and silver layers, external interference was prevented, and the SERS signal was 151.7-fold stronger when compared to GNRs only. Furthermore, this structure provides an outer shell for functional modifications without the traditional competitive binding of Raman reports. Although the sensor required streptavidin-modified magnetic beads as capture substrates, which may be viewed as an extra step compared to previous sensors, it was successfully tested on real blood samples of cancer patients [78].

Multiplexed detection of biomarkers can also be applied in the diagnosis of neurodegenerative diseases, such as Alzheimer’s disease (AD). Zhang et al., engineered a SERS detection platform that detected both Aβ (1–42) oligomers and Tau protein simultaneously [79]. To do so, the team combined different Raman dye-coded polyA aptamers with AuNPs. The simplicity of the system was based on two types of Raman dye-coded polyA aptamer-AuNPs. In the presence of the protein of interest, the polyA aptamer would detach from the AuNPs which would induce the aggregation of the NPs. This would activate a strong SERS signal and a colorimetric change in the solution (Figure 3c,d) [79].

### 2.3. MEF-Based Sensing

Metal-enhanced fluorescence (MEF) is a promising application of aptameric plasmon sensors. As the intensity of MEF can be affected significantly by the distance between the fluorophore and the localized surface plasmon resonance-generating structure, aptamers present a convenient method to optimize such factors [80]. Hence, Choi et al., took advantage of this aspect to design a rapid, simple, and one-step technique for the real-time monitoring of intracellular proteolytic enzymes, such as Caspase-3 (Figure 4a) [81]. The nanobiosensor was based on an AuNP with a double connection to a fluorophore molecule, fluorescein isothiocyanate (FITC). The first link was made of peptides and was meant to keep the FITC molecule close to the AuNP and quenched in the absence of caspase-3. The second link, single-stranded oligonucleotides, was designed to allow an optimal MEF of FITC, in the presence of a proteolytic enzyme that would degrade the peptide connection. The short reaction time of 1 h and the low detection limit of 10 pg/mL allowed for the successful detection of preapoptotic cells [81].

Jiang et al., developed a highly sensitive fluorescence-enhanced aptasensor based on a polyA_n_-aptamer nanostructure for the detection of adenosine triphosphate (ATP) [82]. Their approach involved optimizing the distance between fluorophores, in this case fluorescein (FAM), and the AuNP to enhance fluorescence by adjusting the number of adenosine bases at the 5′ end of a polyA_n_-aptamer. A FAM-bound aptamer would hybridize with the AuNP-bound polyA_n_-aptamer and induce stronger fluorescence. However, in the presence of targeted ATP, the FAM-aptamer would be released, leading to weak fluorescence (Figure 4b). Through this mechanism, a detection limit of 200 pM was successfully achieved when the polyA_n_-aptamer reached a length of 9 nm, a seven-fold sensitivity improvement compared to more common detection methods [82]. Since ATP is a very important molecule in living organisms, it is not surprising to find many studies aiming to improve its detection. Zheng et al., designed a sensing mechanism based on structure-switching aptamer-triggering metal-enhanced fluorescence for Cy7 [83]. Gold nanobipyramids were synthesized by a seed-mediated growth method, then separated and functionalized with an aptamer probe that can bind ATP (Figure 4c,d). The team reported the first hybridization chain reaction-induced MEF, but despite the novelty of their method, the detection limit achieved was not as low as other biosensors with a marked 35 nm [83].

**Figure 4 biosensors-12-00418-f004:**
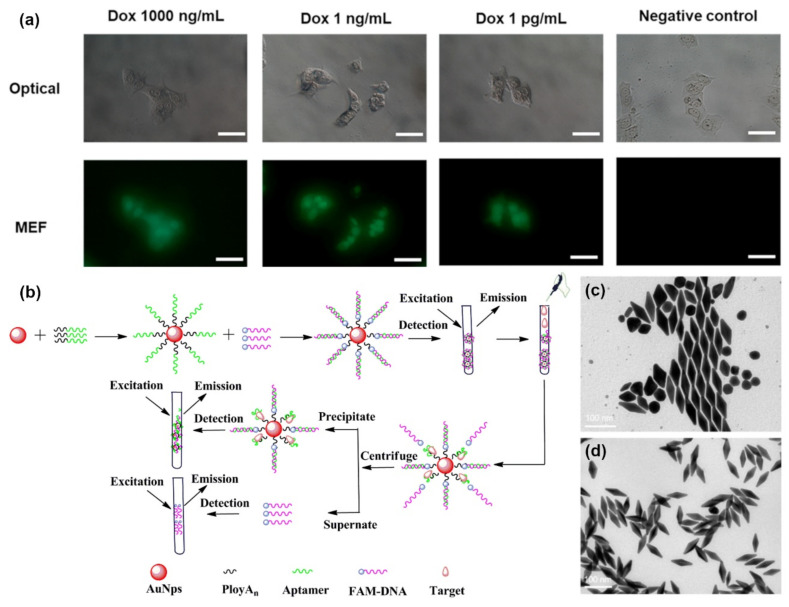
Metal-enhanced fluorescence biosensors: (**a**) Fluorescence imaging of Dox-treated MCF-7 cells. (Scale bars are 100 μm) (**b**) AuNPs@polyAn-aptamer@FAM-DNA nanostructure for ATP detection. (**c**) TEM image of synthetized AuNBPs with LPRW at 751 nm before separation. (**d**) TEM image of AuNBPs with LPRW at 751 nm after separation. (Reproduced with permission from [81], published by American Chemical Society 2020; reproduced with permission from [81], published by Elsevier 2019; reproduced with permission from [83], published by Elsevier 2020).

MEF aptasensors were also proven to be efficient in the detection of viral DNA. Furui et al., constructed another MEF microarray for the detection of hepatitis B with a LOD of 50 fM [84]. The enzyme free detection mechanism was based on the sandwich binding of Tag-A and Tag-B, both aptamer-functionalized silver nanoparticles (AgNPs), through a capture probe immobilized on to the chip with the viral target serving as a bridge. The detection assay demonstrated up to 120-fold enhancement factor due to the increasing number of fluorophores and the MEF caused by the AgNPs [84].

## 3. DNA Origami-Based Plasmonic Biosensor

Due to its versatile structure modifications, DNA origami can easily be applied in the development of biosensors [85,86]. Biosensors designed with synthetic DNA origami are rolled to two specific goals. First, the specific structure of DNA origami was applied to a biosensor platform. For example, the DNA tetrahedron or DNA branch structure can be introduced to bioprobe [85,87]. Second, each arm is designed to have unique functionalities including target recognition, signal generation or immobilization. Conventional single-stranded or double-stranded DNA only has two ends. However, DNA origami, such as Y shape or 3WJ DNA, has an additional arm to serve as a multi-functional bioprobe [53,88]. Furthermore, the X shape or 4WJ can improve functionality significantly. Assembled, multi-functional DNA origami can reduce the fabrication and labeling process for biosensors.

Lee et al., developed the LSPR biosensor which contained a multi-functional DNA 3WJ on the hollow Au spike-like NP-modified substrate for detecting avian influenza [89]. In this report, DNA 3WJ was designed to provide several functionalities, such as target recognition, immobilization, and signal enhancement, simultaneously. Each end of the DNA arm has a functional moiety. The DNA 3WJ-a arm was tagged to the hemagglutinin (HA)-binding aptamer as the bioprobe, the DNA 3WJ-b arm was connected to the FAM dye for signal enhancement function, and the DNA 3WJ-c arm was modified to the thiol group for anchoring the surface. Hollow Au spike-like NPs provided the plasmonic platform for target detection (Figure 5a,b). In this paper, Hollow Au spike-like NPs provided more surface roughness compared to normal Au nanoparticle that can provide more bioprobe immobilization number. Additionally, Hollow Au spike structure enhanced the LSPR effect compared to normal Au NP. Based on this DNA 3WJ and NP combination, the HA protein can be detected in PBS buffer and diluted chicken serum. The detection range of HA protein is 1 pM to 100 nM with high selectivity (Figure 5c,d). Thus, DNA 3WJ is a great tool for LSPR biosensor fabrication.

Puchkova et al., demonstrated fluorescence enhancement using a DNA origami nanoantenna technology and proposed an MEF sensor [90]. The researchers attached 100 nm colloidal gold NPs to columnar DNA origami and a fluorophore was incorporated into the inter-particle spacing. The gold NPs were functionalized with thiol groups and bound to DNA origami to induce a particle distance between 12 and 17 nm. Furthermore, the authors quenched the fluorophore with NiCl_2,_ employing the phenomenon that effective quantum yield improves if the intrinsic quantum yield of the fluorophore is lower (>10). As a result, a fluorescence enhancement index of 306 was obtained, which is the highest value reported to date (Figure 6a,b). In this contribution, the researchers showed substantial improvements in the plasmonic structure by improving DNA origami, reducing the interparticle distance, and introducing a quencher to reduce the fluorophore quantum yield. Since nanoantennas are shown to be capable of detecting individual molecules at concentrations as low as 25 uM, DNA nanotechnology creates the possibility of placing molecules in self-assembled plasmonic nanoantennas.

Thacker et al., showed that DNA origami and two gold NPs are powerful tools for generating SERS active NPs [91]. Strong plasmonic binding was induced by holding two gold NPs (40 nm in diameter) at a distance of 3.3 ± 1 nm, which is one of the shortest intervals that can be made by DNA origami assembly. Moreover, the attachment of individual ssDNA-coated gold NPs to DNA origami sheets allowed for the characterization of the red shift and the subsequent extraction of the effective refractive index of DNA origami. Consequently, they showed the efficiency of dimer structures for SERS measurements using enhancement factors of up to 7000 times. It showed local field enhancement through the detection of short single-stranded DNA oligonucleotides as well as a small number of dye molecules. This demonstrated the effectiveness of the combination of DNA origami and SERS, suggesting that it has great potential in various biosensing fields.

There is also a case where DNA origami became a template and SERS was programmed. Zhou et al., improved the complexity of SERS programming using DNA origami to increase the number of NPs in SERS metamolecules and arrange them into elaborate configurations [92]. The template was designed based on DNA origami hexagon tiles (DHT), and six gold NPs (10 nm in diameter) were fixed on the outer surface of the hexagon, and Ag–Au core−shell NPs were formed through subsequent Ag growth. The longer the silver growth time, the longer the color wavelength, and the metamolecules containing the largest NPs had the strongest interparticle electromagnetic field and Raman enhancement effect (Figure 6c,d). In general, SERS metamolecules are assembled into clusters of a small number (2–4) of nanoparticles and have limited programmability, but in this study, they expanded the structural complexity by increasing the number of nanoparticles. In other words, new plasmon research was made possible using DNA origami templates to precisely control the structural composition of metal NP clusters. This technology, it was shown, has several applications in fields such as photonics and sensing.

Jung et al., confirmed that DNA nanotechnology capable of calculating molecular information, called toehold-mediated DNA strand displacement (TMSD), could be linked to in vitro transcription and act as an information-processing unit for cell-free biosensors [93]. They developed design rules for interfacing small molecule sensing platforms with toehold-mediated strand displacements to construct hybrid RNA-DNA circuits that allow fine-tuning of reaction kinetics. Additionally, a cell-free biosensor platform, called an RNA output sensor, could be activated by ligand induction and combined with the computational power of TMSD to construct 12 different circuits that realize seven logical functions. Thus, a circuit capable of estimating the concentration range of a target compound in the sample was designed and verified. As a result, they extended their functions by interfacing with toehold-mediated strand displacement circuits through programmable interactions between nucleic acid strands. This platform presented the potential for different types of molecular computations in cell-free systems and of TMSD circuits that improve cell-free biosensing technology.

Dass et al., presented recent studies of plasmon sensing using DNA-based nanostructures. Using DNA origami for the coordinated arrangement of plasmonic nanoparticles holds great promise in the field of biosensing. Plasmonic particles can also be detected, ranging from fluorescence enhancement to enabling visualization of nanoscale dynamic behavior. Among prevalent technologies, DNA-based nanotechnology is the most successful strategy, and plasmonics coupled with DNA origami provide a novel and more effective approach [94]. This capability provides a suitable platform for the fabrication of plasmonic sensors for single-molecule measurements. From this point of view, the authors discussed recent developments and future research directions for plasmonic sensing with DNA origami, such as fluorescence-based plasmon sensing, SERS sensing, and chiral sensing. It was concluded that the fabrication of DNA origami-based plasmonic sensors is a new technology platform for engineering molecular scale sensors and has reached a level that can be applied in the field.

Recently, the DNAzyme is also used as the bioprobe for constructing plasmon biosensor. Conventionally, in the area of biosensors, the DNAzyme has been used to construct electrochemical biosensor beause of thier catalytic effect including H_2_O_2_ reduction or DNA cleavage [95,96,97].

Other than aptamer or origami-based plasmonic biosensors, few researchers have worked on developing plasmonic biosensors. Only recently have scientists introduced this unique material for plasmon biosensor fabrication. The group developed a DNAzyme integrated with magnetic beads (MBs) to create an *Escherichia coli* specific plasmonic biosensor [98]. They used DNAzyme because its RNA-cleaving characteristics would offer accurate target recognition. The MBs provide the plasmonic effect for the biosensor. When the DNAzyme-MB reacted with *E. coli* lysate, DNA cleavage was activated. Based on this, signal amplification was processed by a hybridization chain reaction with silver etching via an enzyme-triggered reaction. This detection method can selectively detect *E. coli* compared to other samples including *M. subtilis, M. peli* etc. *E. coli* was even detected in apple juice or skim milk using the DNAzyme-MB system. The detection range of *E. coli* was reportedly 50 to 5 × 106 cfu/mL.

Lee et al., also reported the use of DNAzyme conjugated with gold NPs for detecting *Salmonella choleraesuis* [99]. This study used multi-component DNAzymes for performing various functions in plasmonic biosensors. Multi-component DNAzyme can cleave the target, make the color change, and can be attached with gold NPs for plasmonic detection. The proposed biosensor can detect the 16 rRNA real target sample in the range of 50 nM to 500 nM. Moreover, Wu et al., fabricated a lead (II) ion detecting SPR biosensor composed of DNAzyme-gold NP hybrids [100]. The substrate cleavage characteristic with Pb^+2^ ions using the DNAzyme can be combined with the gold NP for SPR biosensor. LOD was determined to 80 p.m. in drinking water. Thus, DNAzyme is also a versatile bioprobe for plasmonic biosensor development.

## 4. Conclusions

This review covered the most recent progress in nucleic acid-based plasmonic biosensor fabrication (Table 1). From cancer marker sensing to toxin detection, the combination of plasmon resonance-generating structures with oligonucleotides, such as aptamers, or more complex constructions, such as origami folded aptamers and DNAzymes, has certainly proven to be efficient. Not only do nucleic acid-mediated sensors have better affinity and stability, they are also easier to produce. Furthermore, surface plasmon resonance-based sensing presents numerous advantages, such as lower detection limits due to their increased sensitivity and a shorter sensing time. Yet, certain aspects, such as the nonspecific adsorption of unwanted molecules and the production of homogeneous plasmon surfaces should be considered for further improvements. Only then can the potential of plasmonic biosensors for application in real sample analysis be achieved. Nonetheless, based on recent studies, DNA nanotechnology for plasmonic biosensors offers a more sensitive and less costly alternative to other traditional sensing methods.

## Figures and Tables

**Figure 1 biosensors-12-00418-f001:**
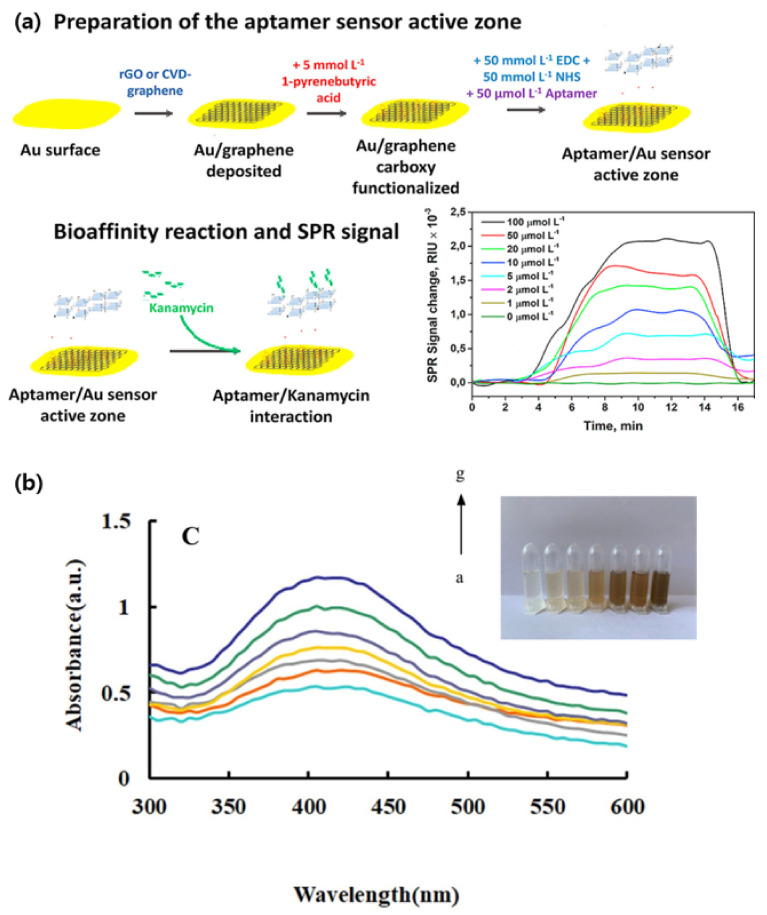
Aptamer-based SPR detection mechanism: (**a**) Kanamycin sensing platform. (**b**) Surface plasmon resonance absorption spectra of the nanosilver sol systems with a–g as As^3+^ concentrations of 0; 0.0945; 0.189; 0.2835; 0.378; 0.4725 and 0.567 μg/L. (Reproduced with permission from [67], published by Elsevier 2021; reproduced with permission from [69], published by Elsevier 2020).

**Figure 2 biosensors-12-00418-f002:**
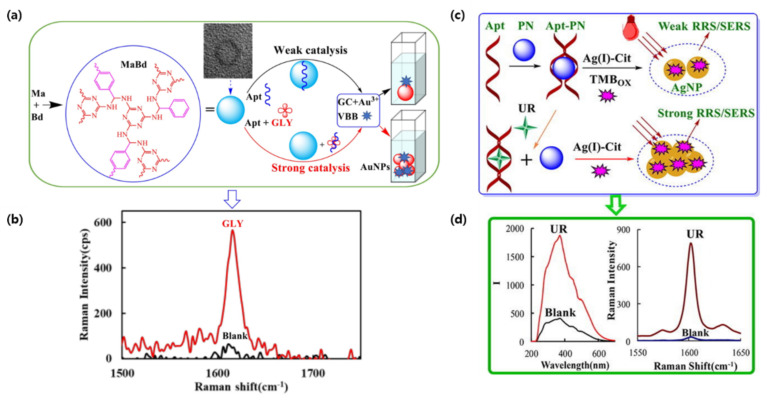
SERS aptameric catalysis mechanism: (**a**) Aptamer-mediated MaBd catalysis of GC-HAuCl4 nanogold reaction for the detection of glyphosate. (**b**) Raman spectra of glyphosate detection peak. (**c**) Aptamer-mediated PN-AgNO_3_-Cit catalytic amplification reaction for the detection of urea. (**d**) Raman spectra of RRS and SERS dual-mode urea detection. (Reproduced with permission from [74], published by Elsevier 2021; reproduced with permission from [76], published by Elsevier 2022).

**Figure 3 biosensors-12-00418-f003:**
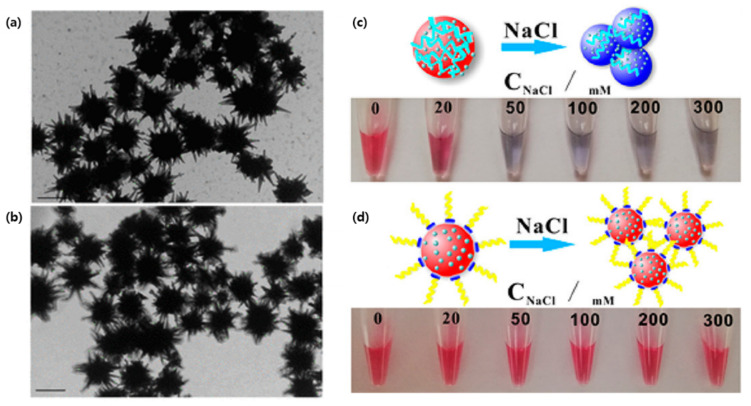
Gold nanoparticles aptamer binding: (**a**) TEM of synthesized gold nanostars and (**b**) 4-aminothiophenol and EpCAM aptamer-functionalized gold nanostars. (Scale bars are 100 nm) (**c**) Photo of the ionic strength-dependent color change of the non-polyA-Tau apt/DTNB/AuNPs. (**d**) Photo of the ionic strength-dependent color change of the polyA-Tau apt/DTNB/AuNPs. (Reproduced with permission from [77], published by American Chemical Society 2018; reproduced with permission from [79], published by American Chemical Society 2019).

**Figure 5 biosensors-12-00418-f005:**
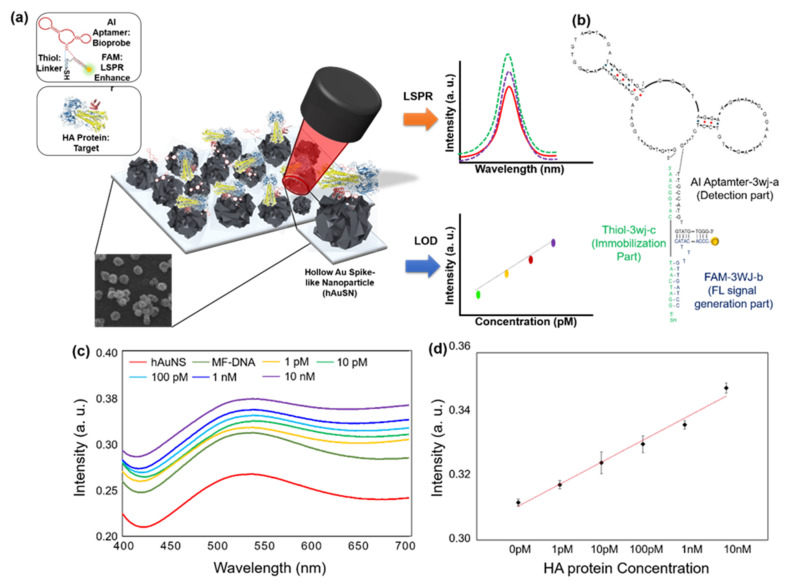
(**a**) Schematic diagram of multi-functional DNA 3WJ for AIV H5N1 detection through LSPR, (**b**) Expected 2D structure of multi-functional DNA 3WJ, HA protein, spike protein. (**c**) Detection of HA protein in 10% chicken serum on the DNA 3WJ-based localized surface plasmon resonance (LSPR) biosensor. Absorbance increases from different HA concentrations in 10% chicken serum (1 pM to 10 nM) of (**d**) Calibration characteristics of the different concentration of HA protein range from 1 pM to 10 nm with correlation coefficient (R2) of 0.9796. (Reproduced with permission from [89], published by Elsevier 2019).

**Figure 6 biosensors-12-00418-f006:**
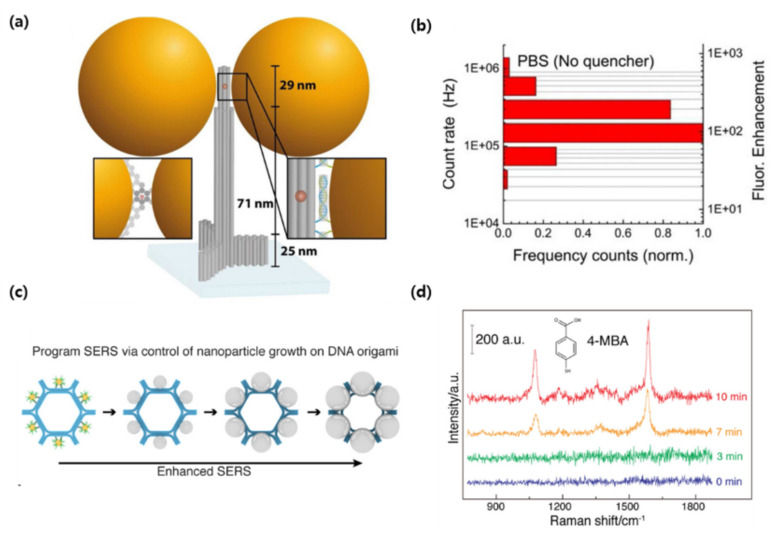
(**a**) Sketch of the DNA origami pillar (gray) employed to build the optical nanoantenna with two 100 nm Au nanoparticles together with a top-view (lower-left inset), (**b**) Photon count rate histogram for the dimer nanoantennas with no quencher, (**c**) Design of DNA origami-templated metamolecules with programmable surface-enhanced Raman scattering (**d**) Single-particle Raman spectra of Raman tag (4-mercaptobenzoic acid, 4-MBA) after adsorbed onto Ag@Au hexagon monomer metamolecules of varied silver growth time. (Reproduced with permission from [90], published by American Chemical Society 2015; reproduced with permission from [92], published by Elsevier 2020).

**Table 1 biosensors-12-00418-t001:** DNA nanotechnology for plasmonic biosensor construction.

Detection Method	Oligonucleotide Component	Plasmonic Component	Target	LOD	Reference
SPR ^1^	Aptamer	AuNPs	Oxytetracycline	0.50 ng/mL	[66]
		Graphene coated Au chip	Kanamycin	285 nmol/L	[67]
		niobium carbide MXene quantum dots coated Au chip	N-gene of SARS-CoV-2	4.9 pg/mL	[68]
		Ag nanosols	Arsenic (As^3+^)	0.01 μg/L	[69]
LSPR ^2^	Aptamer	AuNPs	Ochratoxin A, triphosphate, 17β-estradiol and oxytetracycline hydrochloride	10^−10.5^ g/mL	[71]
	DNA 3 Way Junction	Hollow Au nanospike	Avian influenza	1 pM	[89]
SERS ^3^	Aptamer	Au nanosols	Glyphosate	0.002 nmol/L	[74]
		Ag nanosols	Urea	0.03 nM	[76]
		Au nanostars	Epithelial cell adhesion molecule (EpCAM)	100 nM to 500 nM.	[77]
		Au–Ag bimetallic nanotrepangs	PSMA, Her2 and AFP proteins expressing exosomes derived from LNCaP, SKBR3 and HepG2 cell lines	6 particles/μL, 72 particles/μL and 35 particles/μL, respectively	[78]
		AuNPs	Tau protein and Aβ(1–42)	3.7 × 10^−2^ nM	[79]
	DNA origami	AuNPs	N/A	N/A	[90]
			N/A	N/A	[91]
		AuNPs and AgNPs	N/A	N/A	[92]
MEF ^4^	Aptamer	AuNPs	Caspase-3	10 pg/mL	[81]
			ATP	200 pM	[82]
		Au nanobipyramids	ATP	35 nM	[83]
		AgNPs aggregates	Hepatitis B virus DNA	50 fM	[84]

^1^ Surface Plasmon Resonance. ^2^ Localized Surface Plasmon Resonance. ^3^ Surface-Enhanced Raman Scattering. ^4^ Metal-Enhanced Fluorecence.

## Data Availability

Data sharing not applicable to this article.

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
