# Peer review of "Recent Advances in DNA Nanotechnology for Plasmonic Biosensor Construction"

_biosensors, 2022, doi:10.3390/bios12060418_

Round 1

Reviewer 1 Report

The authors focus this review on the recent progress of nucleic acid engineering for plasmonic biosensor construction. They discussed the applications of both simple aptameric structures and origami-shaped structures in SPR (surface plasmon resonance), SERS (surface enhanced Raman Scattering), and MEF (Metal-enhanced fluorescence). Technically, the manuscript is well written and the results are interesting for the readers. However, the necessary references are absent in the manuscript to explain the obtained results. Therefore, in my opinion, a minor revision is needed to accommodate the high-quality requirements of this Journal. 

1.     Line 173, states “catalysis of AuNPs synthesis”. For the readers to comprehend the experimental setup of plasmonic photocatalytic reactions, a related paper (Hung et al., Catalysts 10 (1), 46 (2019)) is suggested to quote in the text.

2.     Typo, e.g., line 252, in “a polyAn-aptamer “, “n” should be subscript. Line 175, in “1617 cm-1”, “-1” should be superscript. Please check it. In the same manner, please check throughout this manuscript.

3.     The manuscript needs to be proofread in several languages and grammatical errors.

Reviewer 2 Report

The manuscript by Park et al. reviewing recent advances in DNA nanotechnology for plasmonic biosensor construction can provide a useful overview for the readers in this field to refer, if it is complemented with the following points.

1. The title of the manuscript is “Recent advances in DNA nanotechnology for plasmonic biosensor construction” but there is only one short paragraph about DNA nanotechnology. This section should be revised with more profound introduction, especially its advantages and merits compared to other technologies for plasmonic biosensors, to provide background and fundamental to support the readers in comprehending the content of the review.

2. There are only two applications of this technology (cancer marker sensing and toxic detection) reviewed in main content. Are they the only important applications of this technology or are there other ones which should also be considered? Reviewing some other applications can significantly improve the quality of the manuscript.

3. What is the domain of “recent advances” (time period of the publications) reviewed in this manuscript? What are achievements that the publications reviewed in this manuscript contributed to this field compared to the previous (older) ones (for example, the novel trends/fashions in design/architecture and/or any other specifications lead to breakthrough/improvement of (novel) applications)? Specifying and clarifying them in the manuscript can highlight its novelty and distinguish them from the similar reviews on this topic.

4. A table summarizing the publications reviewed in this manuscript with specifications of design/architecture, probe, target, limit of detection, detection range, etc. is essential for the convenience of the readers in scanning the manuscript.

Reviewer 3 Report

The authors of the manuscript “Recent advances in DNA nanotechnology for plasmonic biosensor construction” reviewed recent developments in biosensors based on plasmonic resonance techniques combined with DNA nanotechnology. The topic would be attractive to many researchers, but the contents should be improved.

I have a few questions and suggestions.

1: I am afraid the concept of plasmonics is not correct. And the Raman scattering should be explained by its features.

2: Plasmonic sensors are not label-free since these sensors do not have specificity if the sensing mechanism is based on the refractive index change. Most of the occasions, specific molecules are needed to bind the target molecules.

3: What is the standard method mentioned in the first paragraph on Page 6 for arsenic detection.

4: Line 157 to 162 is confusing.

5: More details of each plasmonic biosensor should be included. Fundamental information should be provided,e.g.how the metallic nanoparticle enhanced the Raman signal.

6:The connections, comparisons, or the development of the DNA techniques should be mentioned in each example to make the structure of the review clear. The transition of the review should be improved.

Round 2

Reviewer 2 Report

1) There is a replicated sentence (lines 93-95 and 122-123) and 2) exchanging the positions of the paragraph introducing plasmonic biosensors (lines 96-117) with the paragraph discussing the merits of incorporating DNA nanotechnology into the plasmonic biosensors (lines 85-95) may improve the fluency of the manuscript. However, it is the authors’ will to revise them since it will not affect the academic content of the manuscript, which has been significantly improved by the authors’ revision.

Reviewer 3 Report

The revised version of “Recent advances in DNA nanotechnology for plasmonic biosensor construction” is well organized with more scientific details provided for the readers. This manuscript is ready for publication.